# New Technology for Production of Dicyclopentadiene and Methyl-Dicyklopentadiene

**DOI:** 10.3390/polym14040667

**Published:** 2022-02-10

**Authors:** Tomáš Herink, Petr Fulín, Jiří Krupka, Josef Pašek

**Affiliations:** 1ORLEN Unipetrol RPA, Záluží 1, 436 70 Litvinov, Czech Republic; tomas-herink@seznam.cz; 2Faculty of Chemical Technology, University of Chemistry and Technology, Technická 5, 166 28 Prague, Czech Republic; jiri.krupka@vscht.cz (J.K.); josef.pasek@vscht.cz (J.P.)

**Keywords:** dicyclopentadiene, unsaturated polyester resins, cyclic olefin copolymers

## Abstract

ORLEN Unipetrol’s Steam Cracking unit processes a wide range of hydrocarbons from gases to heavy oils produced in refinery processes. Due to the heavy feedstock, the Steam Cracking unit can produce very valuable hydrocarbons such as cyclopentadiene and dicyclopentadiene in addition to ethylene, propylene and benzene. These hydrocarbons can be obtained and used as very profitable monomers for many other chemical applications. ORLEN Unipetrpol, in cooperation with the University of Chemistry and Technology in Prague, has developed a technology for the isolation of technical dicyclopentadiene of both medium purity grades and high purity grades. Making DCPD grades will add considerable value to the raw C5 by-product stream from the Steam Cracker pyrolysis gasoline. The capacity of the new existing DCPD unit is expected to be in the range of 20–26 thousand metric tons per year, depending on the derivative product portfolio and purity of the DCPD. The construction of the unit started in September 2020, and production is expected to be launched in the second half of 2022.

## 1. Introduction

The cracking process of hydrocarbon mixtures produces a number of hydrocarbons important for industrial chemistry, including hydrocarbons such as isoprene, piperylene and cyclopentadiene (CPD), respectively, and its dimer, dicyclopentadiene (DCPD). All these dienes are very attractive from the point of view of further chemical use, as they represent monomers for a number of different applications.

Dicyclopentadiene (DCPD) is formed very easily by the dimerization of 1,3-cyclopentadiene (CPD) by the Diels–Alder mechanism. The reaction is strongly exothermic and takes place spontaneously, even at temperatures around 20 °C. At higher temperatures, dicyclopentadiene can be converted back to the monomer.

Both compounds form an interconnected system that can be used in a number of chemical reactions and industrial technologies. Due to the chemical reactivity of the DCPD/CPD system, dicyclopentadiene is a key raw material for an important group of related monomers, polymers, resins and a number of high added-value specialties representing pharmaceuticals and perfume components [1]. The dicyclopentadiene can be used in oil refining as an additive for diesel fuel improving its properties such as a cetane number or sooting tendency [2].

Dicyclopentadiene exists in two stereoisomeric forms: exo and endo. The product of CPD dimerization at temperatures up to about 150 °C is almost exclusively endo-DCPD, while the exo isomer is formed only at higher temperatures [3]. Commercial DCPD products contain mostly endo-DCPD only. Chemically pure DCPD is a colorless crystalline substance with a melting point of 34 °C and a boiling point of 172 °C, but lower purity commercial DCPD is mostly liquid.

Cyclopentadiene is a colorless liquid with a boiling point of 41 °C. It can be practically stored only at very low temperatures, where the dimerization reaction is slow therefore this solution is not used, and all cyclopentadiene is converted to a more stable dimer. It is stored and transported in the form of a DCPD dimer. In chemical reactions or technologies, DCPD is the starting raw material, although in some cases, the reactant is monomeric cyclopentadiene; an example can be the production of ethylidenenorbornene for EPDM elastomers.

## 2. Dicyclopentadiene Uses

The only significant industrial source of cyclopentadiene and dicyclopentadiene is currently the process of pyrolysis of hydrocarbon mixtures. There are two general categories of industrial end uses of cyclopentadiene and dicyclopentadiene:A.Resins and polymers including hydrocarbon resins (HCR), unsaturated polyester resins (UPR) and ethylene propylene diene rubbers (EPDM);B.Special polymers and fine chemicals including cyclic olefin copolymers (COC), flame retardants, agrochemicals, specialty norbornenes, flavor and fragrance intermediates.

DCPD is sold in several purity grades. Lower purity grade in the 60 wt. % to 80 wt. % range is used in the production of hydrocarbon resins [4]. Grades with a concentration above 85 wt. % are used for the production of unsaturated polyester resins. A high purity grade typically contains 91–95 wt. % DCPD. These grades are used as a source of CPD through DCPD cracking to produce intermediates such as norbornene or ethylidene norbornene (ENB). ENB and high purity DCPD are used in the production of ethylene propylene diene rubbers (EPDM). If DCPD is used as a thermonomer for EPDM, then it tends to be the highest purity, more than 94 wt. %. High purity DCPD can be further upgraded to an ultra-pure grade of over 99 wt. % DCPD. This grade of DCPD is used to produce poly-DCPD for use in catalytic reaction injection molding (RIM).

Hydrocarbon resins are formed by oligomerization (thermal, catalytic) of dicyclopentadiene and other components contained in the raw material, such as co-dimers CPD-isoprene, CPD-piperylene or isomers of methyl-dicyclopentadiene or alternative polymeric-based triglycerides and monomers, such as palm oil-based polymeric materials [5]. Sometimes the resins are hydrogenated to improve color (water-free resin) and thermal stability. Unsaturated DCPD resins are used, namely for coatings production. They can react with selected chemicals such as maleic anhydride to improve mechanical properties. Hydrogenated DCPD resins are used as hot melt adhesives with thermoplastic rubbers, adhesives for diapers and sanitary ware, tackifiers in tires and floor tiles, surface coating and varnishes and ink. They are also used in white for road marking paint, where their light stability and chemical resistance are important. Chemically modified DCPD resins and DCPD copolymers are used in printing inks, especially for offset printing in the newspaper industry.

Unsaturated polyester resins [6] are produced by polycondensation of saturated and unsaturated dicarboxylic acids with polyhydric alcohols, especially glycols. These materials are used in construction in the shipbuilding or automotive industries, primarily in the fiber-reinforced form as laminates, castings and coatings. There are also alternatives to petroleum-based UPR resources such as fatty acid monomers [7]. DCPD was originally introduced into the production of polyester resins as a cheaper substitute for phthalic anhydride for lower quality unsaturated polyester resins [8,9,10]. These types can have the significant advantage of lower costs due to the lower need for styrene and the replacement of phthalic anhydride and propylene glycol by DCPD and ethylene glycol. DCPD-based unsaturated polyester resins have their advantages, particularly in lower viscosity and better processability, better surface properties and better resistance to weathering and water. These DCPD-containing resins are not suitable for applications where transparency is required.

The reaction of cyclopentadiene (CPD) with olefins and dienes produces a number of norbornenes. CPD is developed in situ by the decomposition of DCPD. The most commonly commercially produced is ethylidene norbornene [11,12], which is produced by reacting CPD with 1,3-butadiene and is used as the preferred thermonomer for the production of EPDM elastomer [13]. With the continuing development of copolymers of olefins with cycloolefins, norbornenes find more significant applications in modifying the properties of these new and interesting polymers.

Copolymers with cycloolefins [14] are amorphous transparent polymers similar to polycarbonates and polymethyl methacrylate. Cyclic olefin copolymers are produced by the copolymerization of ethylene with a cyclic olefin monomer such as norbornene [15], dihydro DCPD, phenyl norbornene and tetracyclododecene [16,17], while Ziegler–Natta and metallocene catalysts are used in the polymerization processes. COC properties are low density, having excellent transparency, low water capillarity, chemical resistance, insulating properties and good processability. The main target market for copolymers with cycloolefins is the production of transparent materials, optics or electronic media such as CDs and CD ROMs, medical labware, high-performance films and pharmaceutical blister packaging. These materials will be competing materials for polycarbonates and possibly polymethyl methacrylate.

DCPD of very high purity (poly-DCPD) is mainly used for the preparation of so-called formulated mixtures [18], which are used in the reactive injection. The resulting solid objects have good impact resistance in a wide range of temperatures. It is used to make intricately formed components, e.g., for vehicles. Ultra-pure DCPD in polymer grade or in very high purity is usually obtained by monomerization of DCPD of lower purity and subsequent dimerization of the resulting cyclopentadiene.

Examples of the use of dicyclopentadiene of varying purity for various applications in construction, automotive, electronics, etc., are shown in Figure 1. Global demand for DCPD in 2017 was at the level of 811 kt, where almost 58% of this demand was located in the Far East region. What is more, it is expected that this region will grow much faster in comparison to other regions, such as Europe or North America. The demand in Europe in 2017 was at the level of 125 kt. The demand is expected to increase by a further 20 per cent by 2030 in Europe, 40 per cent in the US markets and even 60 per cent in Asian markets.

## 3. DCPD Production in ORLEN Unipetrol

The Steam Cracking unit of ORLEN Unipetrol in Litvínov is an important European producer of monomers. The Steam cracking unit treats a wide spectrum of hydrocarbons from gases to heavy oils produced in refinery processes. Due to heavy feedstocks, the Steam Cracking unit can produce, besides ethylene, propylene and benzene, very valuable hydrocarbons such as cyclopentadiene and dicyclopentadiene. In 2020 ORLEN Unipetrol launched the construction of a new unit to produce dicyclopentadiene in a wide range of commercial qualities.

The technology for the isolation of technical DCPD was developed by ORLEN Unipetrol in cooperation with the Institute of Chemical Technology in Prague. There are two main principles by which DCPD can be separated from the pyrolysis gasoline, depending on whether the pyrolysis gasoline is processed immediately once leaving the debutanizer. When the pyrolysis gasoline is processed immediately, the amount of DCPD present is small. The C5 fraction of the pyrolysis gasoline is first separated as the overhead product of a depentanizer. The C5 stream is fed into a dimerizer reactor to convert CPD into DCPD. The separation of the not converted C5 hydrocarbons from the DCPD is secured in the distillation column in which the DCPD is recovered as the bottom product. Another principle is that the pyrolysis gasoline is first dimerized to convert the maximum CPD into DCPD, and then the individual fractions are separated on distillation columns. High-purity DCPD in the range of 98–99% is produced by the monomerization of the lower-purity DCPD. Pure CPD is separated from the crude DCPD and re-dimerized in controlled conditions to avoid oligomer formation. Unreacted CPD is stripped from the high-purity DCPD by mild fractionation.

The developed technology assumes the processing of light pyrolysis gasoline in a series of four distillation columns with a cascade of dimerization reactors. In the Steam Cracking unit, light pyrolysis gasoline is the bottom product of the debutanizer, where C4 hydrocarbons are separated from the mixture of hydrocarbons [19]. The pyrolysis gasoline as a distillation residue consists of a mixture of more than 130 components, mainly C5 and C6 hydrocarbons with a content of about 35 wt. % benzene and 5 wt. % toluene [20,21]. In the debutanizer boiler and subsequently, in the reactors, the CPD contained in the raw material is dimerized. As a result of diene dimerization, pyrolysis gasoline also contains dimers of CPD and MeCPD and the corresponding codimers of these cyclic dienes with linear dienes (isoprene, piperylene).

The process is flexibly designed to produce either 80% or 93–95% DCPD in campaigns. In the case of the isolation of DCPD 80% to 85%, the main impurities are the isomers of methyl dicyclopentadiene (MeDCPD). In reality, it is possible to produce up to 26 kt of DCPD with a concentration of 80% and approximately 20 kt of DCPD with a concentration of 94% per year. A simplified technological diagram of the DCPD production line is shown in Figure 2.

The light pyrolysis gasoline (LPyGas) is the feedstock for DCPD production. Firstly, LPyGas is fed into a dimerizer (R) to convert the remaining CPD to DCPD. The effluent from the dimerizer is fractionated in the first two columns (C1 and C2), where the C5–C9 hydrocarbons are separated from the C10 and heavier fraction. The C10 and heavier stream, which contains 50–70% DCPD, is distilled to yield DCPD as an overhead product of the third column (C3). Finally, the DCPD concentrate is fed to the fourth column (C4) to reach a quality related to color, trimer or peroxides content, etc. All streams which are not processed into DCPD are fed back to the steam cracker. From the diagram, it can be seen that methyldicyclopentadiene (MeDCPD) could be produced as a byproduct. MeDCPD is a very valuable hydrocarbon similar in price and properties to DCPD.

The process has the advantage of flexibility because the purity of DCPD can be produced in the range from 78 to 95 wt. %. It is carried out by adjusting the fractionation conditions in the last two columns (C3 and C4). The quality of both DCPD grades is listed in the Table 1 below.

A valuable concentrate of methyl dicyclopentadiene isomers (MeDCPD) can be advantageously coupled to the implemented dicyclopentadiene technology [22]. The technology for isolating methyl-dicyclopentadiene as a by-product of dicyclopentadiene production was already proposed and is being prepared. It can be assumed the production of methyldicyclopentadiene will take place shortly after the start of production of dicyclopentadiene by 2024 at the latest.

## 4. Conclusions

ORLEN Unipetrol will be able to produce DCPD in the quality of 80 to 94% after putting the unit into operation. The product will have a wide range of uses in the automotive industry, construction, electrical engineering or medicine and pharmacy. The product is in high demand in global markets. There is currently a shortage of DCPD production capacity in Europe, and demand is expected to increase by a further 20% by 2030, by 40% in the US markets and even by 60% in the Asian markets. The investment in the construction of a new production unit amounts to 38 million USD. Completion of the implementation is expected in the first half of 2022, with an installed capacity of 26 thousand tonnes of DCPD per year. The capacity will represent approximately 25% of total production in Europe, with other producers being Dow Chemical (Netherlands), ExxonMobil (France), Shell (Netherlands) and Versalis (Italy). The dicyclopentadiene product is expected to be launched in the second half of 2022.

## Figures and Tables

**Figure 1 polymers-14-00667-f001:**
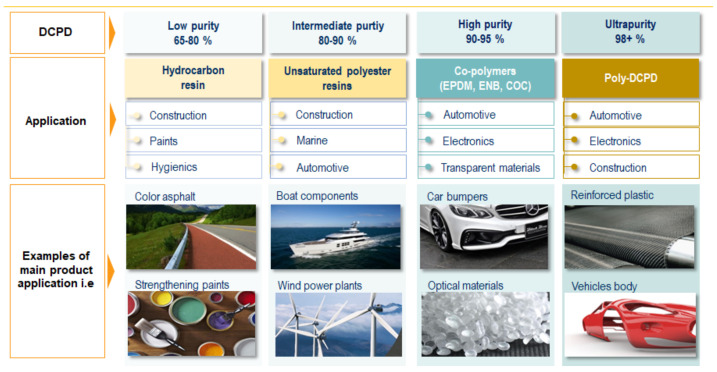
DCPD commercial grades, top consuming sectors and applications.

**Figure 2 polymers-14-00667-f002:**
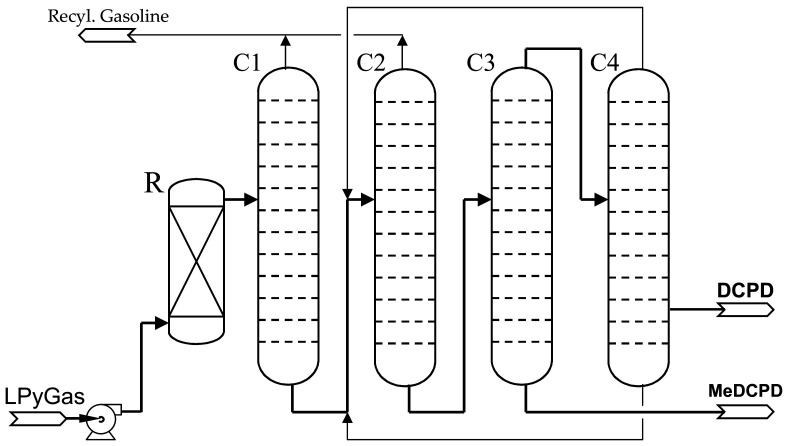
Process scheme for DCPD recovery from light pyrolysis gasoline (LPyGas).

**Table 1 polymers-14-00667-t001:** Dicyclopentadiene product quality.

	Unit	DCPD 80	DCPD 85	DCPD 94	Test
DCPD	wt. %	min 78	min 85	min 93	GC
CPD	wt. %	max 0.05	max 0.05	max 0.05	GC
DCPD + total Dimers	wt. %	min 99	min 99	min 99	GC
Trimers	wt. %	max 0.10	max 0.10	max 0.05	GC
Benzene	ppm	max 1	max 1	max 1	GC
Toluene	ppm	50 max	50 max	50 max	GC
Color, APHA	-	max 100	max 100	max 50	ASTM D 1209
Inhibitor (BHT)	ppm	100–200	100–200	100–200	GC

## Data Availability

Not applicable.

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
