# Peer review of "New Technology for Production of Dicyclopentadiene and Methyl-Dicyklopentadiene"

_polymers, 2022, doi:10.3390/polym14040667_

Round 1
Reviewer 1 Report
The subject of the article is interesting and the manuscript is well written. The manuscript may be recommended for publication in its current form.Author Response
Dear Editors,
We would like to thank very much to Reviewer 1 for careful reading and recommendation for publication in its current form.
Thank you
Sincerely
Tomas Herink

Reviewer 2 Report
Dear Authors,
I have two remarks
1) The reaction figure on page 1 has a wrong sign. I think it is °C. But I see a chinese sign.
2) The table 1 is not readable. You wirite in the first row unit wt%. But then you write min.. Please improve the table i.e. 80 ( after 78 min).
BR
The reviewer
Author Response
Dear Editors,
We would like to thank very much to Reviewer 2 for careful reading and valuable comments helping the manuscript to improve. Please find bellow our responses to comments and recommendations.
Thank you
Sincerely
Tomas Herink
Reviewer 2
1) The reaction figure on page 1 has a wrong sign. I think it is °C. But I see a chinese sign.
It was corrected, there is no Chinese characters in revised manuscript.
2) The table 1 is not readable. You wirite in the first row unit wt%. But then you write min.. Please improve the table i.e. 80 ( after 78 min).
Table was improved.

This manuscript is a resubmission of an earlier submission. The following is a list of the peer review reports and author responses from that submission.
Round 1
Reviewer 1 Report
My opinion is that the paper can be accepted in its current form without substantial changes. However, I ask the authors to review the reversible equation on page 1, where some Chinese characters appeared, at least on the copy that reached to me.
Reviewer 2 Report
This manuscript looks like a technical report. DCPD production in ORLEN Unipetrol does not include satisfactory information for the readers. The quality of presented work is very low. Hence, I would reject the current version and suggest a resubmission.
Reviewer 3 Report
The present manuscript needs to be modified thoroughly. An additional explanation is required.
- Author should provide the comparison of present method with other reported methods.
- Title should be revised as it confuses the readers. Is it review or research paper?
- Comparison data with other monomers should be provided. How present materials are better than others.
- Author should abbreviate the terms used in table 1.
- Author should represent schematically how much DCPD used regionwise?